evolution/molecular biology/genetics

mutation rate, sequence context, human genetics, heterozygosity

**Author for correspondence:**
William Amos
e-mail: w.amos@zoo.cam.ac.uk

# Flanking heterozygosity influences the relative probability of different base substitutions in humans

## William Amos

Department of Zoology, University of Cambridge, Downing Street, Cambridge CB2 3EJ, UK

(iD) WA, 0000-0002-0971-9914

Understanding when, where and which mutations are mostly likely to occur impacts many areas of evolutionary biology, from genetic diseases to phylogenetic reconstruction. Africans and non-African humans differ in the mutability of different triplet base combinations. Africans and non-Africans also differ in mutation rate, possibly because heterozygosity is mutagenic, such that diversity lost when humans expanded out of Africa also lowered the mutation rate. I show that these phenomena are linked: as flanking heterozygosity increases, some triplets become progressively more mutable while others become less so. Africans and non-African show near-identical patterns of dependence on heterozygosity. Thus, the striking differences in triplet mutation frequency between Africans and non-Africans, at least in part, seem to be an emergent property, driven by the way changes in heterozygosity 'out of Africa' have differentially impacted the mutability of different triplets. As heterozygosity decreased, the mutation spectrum outside Africa became enriched for triplet mutations that are favoured by low heterozygosity while those favoured by high heterozygosity became relatively rarer.

## 1. Introduction

Mutations are classically viewed as occurring more or less randomly, to the extent that they form a molecular clock. However, a burgeoning wealth of sequence data has allowed ever-more detailed analyses of factors that impact when, where and what mutations are most likely. As a result, many deviations from the random model have been uncovered [1,2], including variation in rate between related lineages [3], mutation clustering [4,5], higher mutation rates in late-replicating DNA [4] and co-occurrence of mutations on the same strand [4,6]. There is also evidence tying some variation in mutation rate to variation in DNA mismatch repair genes [7]. Despite this, many aspects remain obscure [8].

Two particularly interesting observations show that the mutation process differs between human populations. First, Africans and non-Africans differ in the mutability of different base triplets [9–11]. For example, TCC → TTC mutations are significantly commoner outside Africa compared with inside and, overall, a principal components analysis performed on all 192 possible changes clearly splits Africans from non-Africans and even major geographical regions within Eurasia [10]. More recent analyses have extended these observations to wider genomic contexts and other population groups [9]. Second, Africans and non-Africans appear to differ in their mutation rate, though different datasets yield patterns that run in opposing directions. Over all variants, mutation rate appears slightly higher outside Africa [12], but when based only on variants that probably arose after the 'out of Africa' event, the meaningful part of their history when they became separate populations, the rate is considerably higher in Africa. Across the genome, the difference in mutation rate between Africans and non-Africans is strongly predicted by the amount of variability lost during the 'out of Africa' bottleneck: regions where most variability was lost show the biggest excess mutation rate in Africa [13].

One possible explanation for the link between heterozygosity and mutation rate is that heterozygosity is mutagenic (the heterozygote instability hypothesis, HI) [14–16]. This hypothesis is rooted in observations of meiosis where, in the synaptanemal complex, extensive regions of heteroduplex DNA are formed in which heterozygous sites become mismatches [17,18]. Such mismatches are recognized and 'repaired' by gene conversion-like events [19,20], a process that has been documented in great detail in yeast and probably operates across all diploid eukaryotes. Consequently, DNA surrounding heterozygous sites will tend to experience an extra round of DNA replication in which additional mutations can occur [16], just as it does for DNA replication associated with recombination [21] and as is suggested by the way mutation rate increases near micro-deletions [22]. Taken together, this evidence lends credibility to the idea that heterozygosity and mutation rate could be positively correlated.

The presence of two unexpected patterns, a difference in rate and a difference in type, both seen most strikingly when modern African and non-African humans are compared, begs the question whether the two might be linked. Specifically, since mutation rate appears to correlate with heterozygosity [13,23], could it be that local heterozygosity also impacts the types of mutation that are most likely? Here I test this possibility and show that heterozygosity indeed has a significant impact on the mutability of different triplets.

## 2. Results and discussion

There are two possible approaches for exploring a link between flanking heterozygosity and triplet mutation probability. Most direct would be to analyse large numbers of deeply sequenced mother–father–offspring trios. Unfortunately, the amount of data required still exceeds that which is readily available in the public domain: even the approximately 17 000 mutations identified in the huge Icelandic dataset [4] is modest in the context of identifying subtle trends across 192 possible triplet mutation types and of course does not include African data. The alternative is to use population samples and exploit the fact that the overwhelming majority of rare variants reflect recent mutations. Variants found only in one population are likely to have originated in that population in a context that can be approximated by modern heterozygosity. Although indirect, this approach benefits from the vastly greater number of variants that can be brought into the analysis. I therefore chose to analyse the 1000 genomes Phase 3 data [24].

To obtain the best match between current heterozygosity and heterozygosity when a variant arose, it is vital to focus on the youngest possible variants. It has been estimated that most variants present in just two copies in the 1000 g data are less than 500 generations old [25]. Singletons will be younger still but are more likely to include some unknown proportion of sequencing errors. By contrast, tripletons will be both rarer and, on average, older. In view of this, I decided to focus mainly on singletons and doubletons, but also extracted data for variants present in up to five copies. To reduce further the chance of including older variants, I also required that all copies of a given variant were found in the same major geographical region (Africa, Europe, South Central Asia, East Asia, America).

For each variant, the major alleles and immediate flanking reference sequence bases were used to define a mutating triplet. Each triplet on one strand is equivalent to a complementary triplet on the other strand meaning that the 64 possible triplets can be collapsed to 32 triplets with either A or C as the central mutating base, giving 96 possible mutations (32 triplets times three possible changes for each). Heterozygosity was quantified for a 2 kb window centred on the mutating base, a value chosen to reflect the likely size of gene conversion events [19]. For comparison, I repeated all analyses for a 10-fold larger, 20 kb window. Perhaps not surprisingly given the high levels of linkage disequilibrium at this genomic resolution, the results were essentially identical, so only the 2 kb window results are presented. The

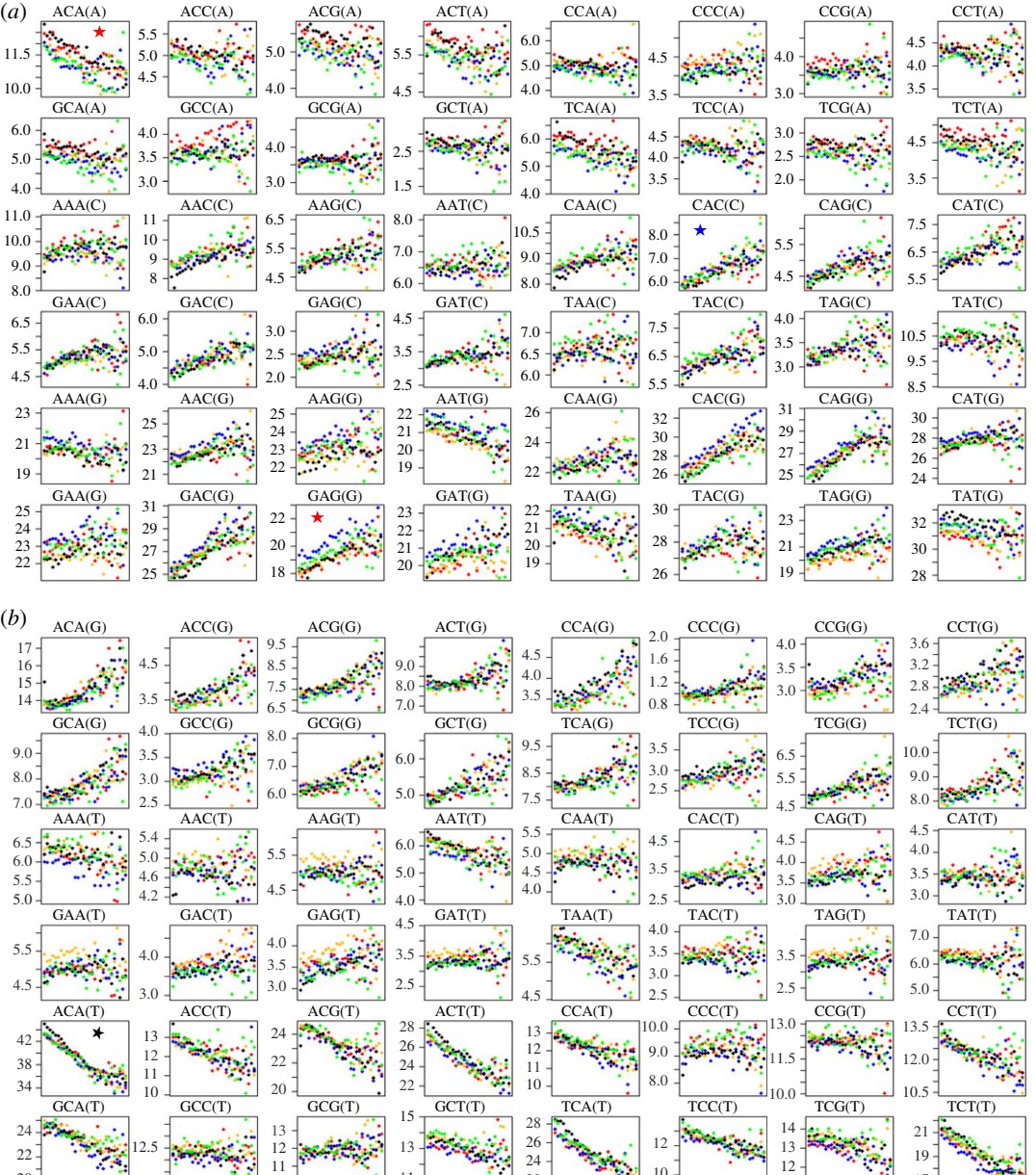

**Figure 1.** (*a,b*) Relationship between heterozygosity and mutating triplet frequency for all 96 possible triplet mutations partitioned by population. Heterozygosity (horizontal axis) is calculated for a 2 kb window centred on each singleton variant in the 1000 genomes data and is allocated to 30 equal bins. Bin 30 contains all sites with scores greater than 30. Vertical axis is proportion of all mutating triplets × 1000. Each panel depicts the data for one type of mutation, given above the panel. ACA(A) signifies mutations of type ACA → AAA. Data for each major geographical region are coded: Africa (black), Europe (red), Southern Asia (blue), East Asia (yellow), America (green). Panels of particular interest are indicated with stars: geographical regions differ (red star); an example of an increasing trend (blue star); an example of a decreasing trend (black star).

1000 g data are low coverage and include much imputation, making individual heterozygosity somewhat unreliable. However, population base frequencies are determined with reasonable accuracy. I therefore calculated expected heterozygosity assuming Hardy–Weinberg proportions and unlinked loci. In reality, variable bases will be in strong linkage disequilibrium, but this should act only to increase the error variance, making any trends found conservative.

General linear models were fitted for each triplet–major geographical region (Africa, Europe, Central South Asia, East Asia, America) combination, with the proportion of all mutating triplets fitted as the binomial response and heterozygosity in 30 discrete bins as the predictor. The data are summarized in 96 individual plots of triplet proportion as predicted by heterozygosity bin (figure 1*a,b*). Many triplets reveal rather weak trends but some either become much more likely (e.g. CAC → C, blue star figure 1*a*) or much

**4**

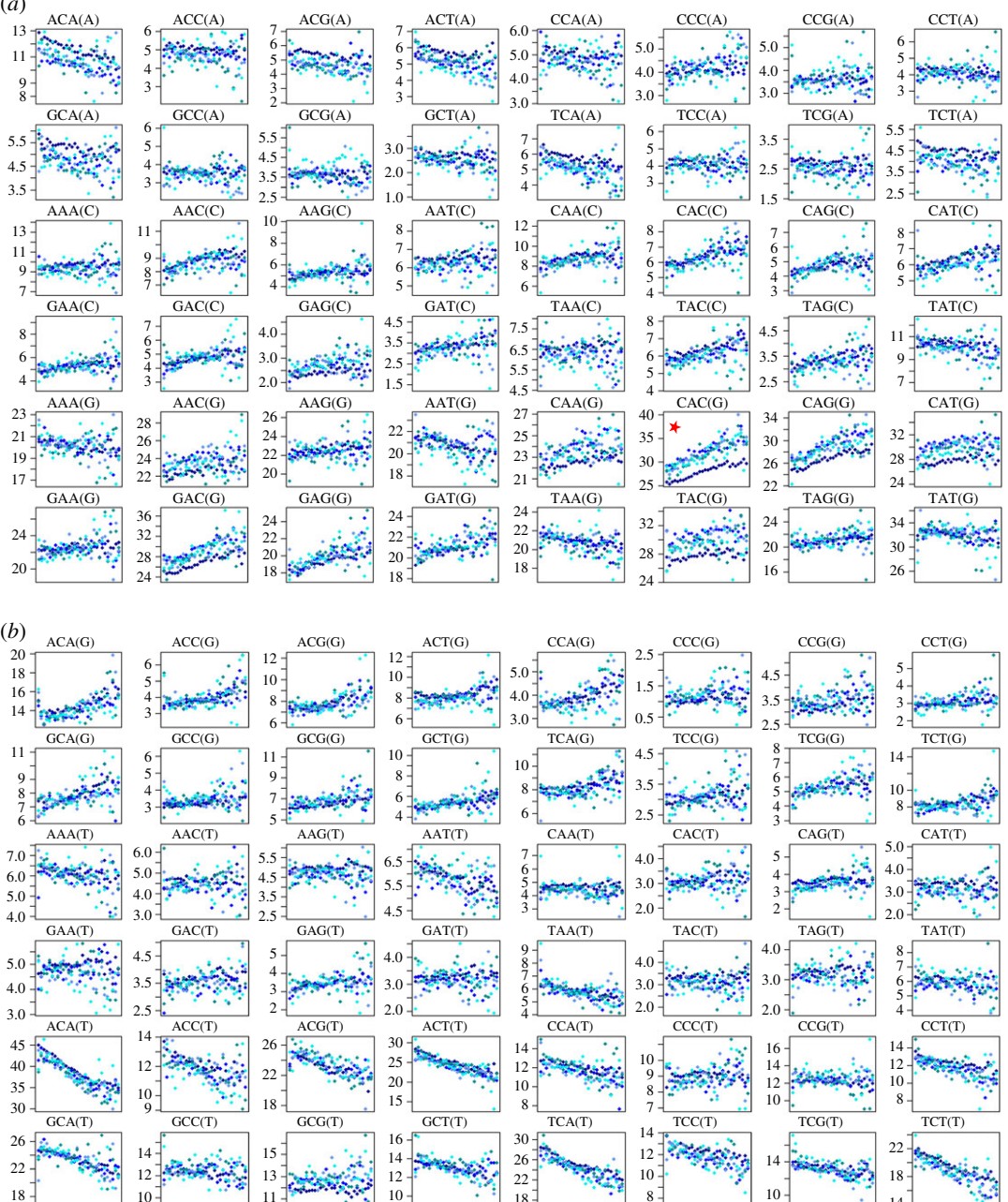

**Figure 2.** (a,b) Relationship between heterozygosity and mutating triplet frequency for all 96 possible triplet mutations partitioned by variant frequency. In these plots, region is held constant (=Africa) and the number of each variant is varied between one and five, with one (singletons) being the darkest blue and five being the lightest. The red star draws attention to the triplet mutation type, CAC(G), where singletons differ most from commoner variants. Other details are given in the legend to figure 1a,b.

less likely (e.g. ACA → T, black star figure 1b) as heterozygosity increases. Effect sizes are modest: in these two examples, the proportions of all mutations that are CAC → C and ACA → T rise and fall, respectively, by 30% and 23% between genomic regions with the lowest and highest heterozygosity. In most cases, all five geographical regions exhibit very similar trends but in a few cases one or more regions differ noticeably in intercept, two examples being indicated with red stars in figure 1a. Interestingly, TCC-T, the triplet showing the biggest African–non-African difference in prevalence [9,11], shows little difference between regions. Harris and Pritchard speculate that the difference between Africa and Europe might be due to a genetic modifier of mutation rate [10,11], and this explanation remains plausible given that the dependency on heterozygosity of this triplet is no stronger than is seen for a number of other triplets.

I also set region to Africa and plotted data for variants present in one (darkest blue) to five (lightest blue) copies (figure 2a,b). Africa was chosen because, with generally more rare variants, there is less scatter. Other

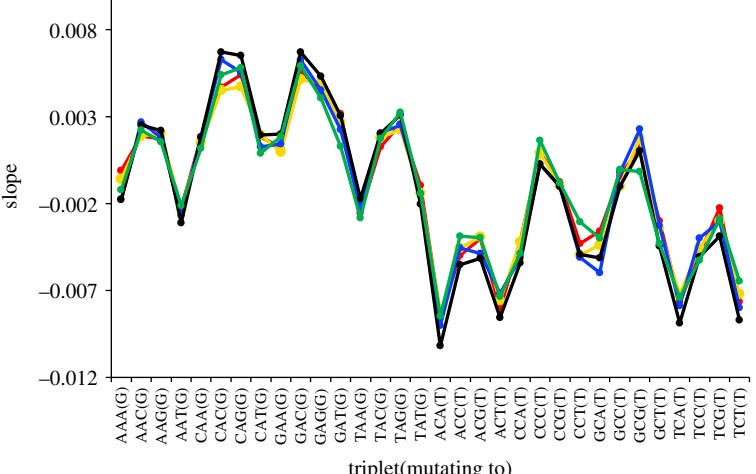

**Figure 3.** Slopes of relationships between flanking sequence heterozygosity and the proportion of different triplets that have recently experienced a transition mutation. Slopes were estimated using binomial regressions and plotted separately for each of the five major geographical regions (see figure 1a for colour key). This plot depicts data for the most abundant variant number class, singletons. Higher numbers of variants yield similar plots but with progressively more variation in the slope estimates. For clarity, data for each region are linked by lines, though the lines themselves have no meaning.

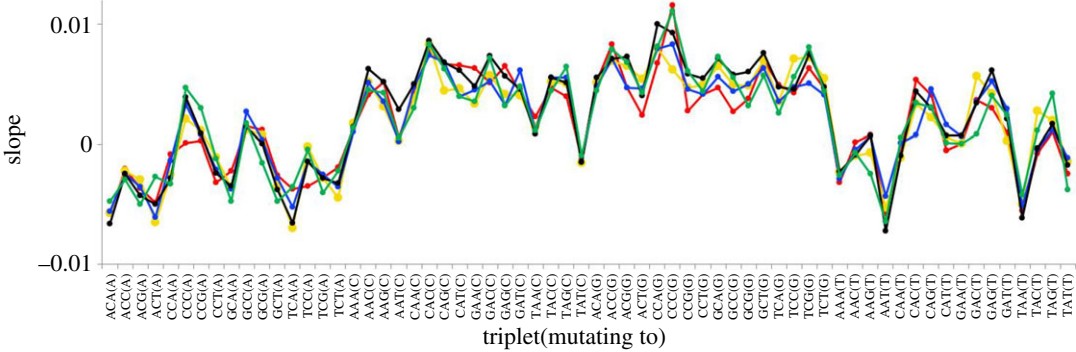

**Figure 4.** Slopes of relationships between flanking sequence heterozygosity and the proportion of different triplets that have recently experienced a transversion mutation. See legend figure 3 for details.

regions give essentially identical results but with more scatter. For most triplets, variants present in different numbers yield very similar trends. However, there is a notable exception, CAC → G, indicated by a red star in figure 2b, where the singletons tend to be consistently relatively rarer than commoner variants. Similar but slightly weaker trends are also observed for other triplets of the form XA[C/G](G), suggesting a role for a CpG effect [2], linked either to sequencing errors or back-mutations.

The various possible triplet changes reveal several higher-order patterns. First, while most C → T transitions increase in relative frequency with increasing heterozygosity, most A → G transitions show a significant decrease. Second, the two possible types of transition mutation generally reveal opposing slopes with heterozygosity. Thus, C → T transitions have mostly negative slopes while A → G transitions are mostly positive. These patterns are illustrated in terms of the actual regression coefficients in figure 3 (transitions) and figure 4 (transversions). Crucially, for any give triplet, all five geographical regions yield extremely similar slopes, particularly for transitions, where sample sizes are larger. Such consistency across independent samples from different geographical regions lends strong support to the idea that, although the effect sizes are small, the trends are highly significant.

In her work on the overall relative frequencies of different mutations, Kelley Harris found a large difference between African and non-African populations [11]. However, when analysed by heterozygosity, the African data points (black) do not stand out as different from any of the other four data series. Consequently, it seems that all populations show extremely similar if not identical dependencies on heterozygosity. The level of similarity is particularly striking given that each variant number equates to a

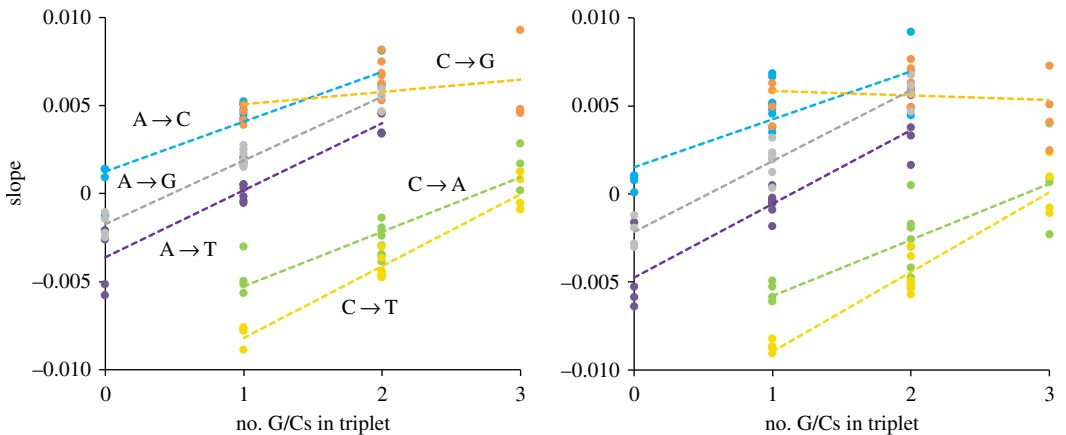

**Figure 5.** Impact of GC content on the relationship between local heterozygosity and triplet mutability. Data are the same as presented in figures 3 and 4, averaged across geographical regions and plotted according to the number of C/Gs in each mutating triplet. Each mutation type is plotted in a different colour, as indicated, and has a separate line fitted. The left-hand panel is for singleton variants, the right-hand panel is for doubletons.

different mean age in each of the different populations [25] and that the heterozygosity surrounding a site where and when a mutation occurred in an individual is only crudely approximated by population level heterozygosity measured in current samples. With such similar dependencies of triplet mutability on heterozygosity, the large loss of heterozygosity 'out of Africa' must have impacted the mutation spectrum and may well go a long way towards explaining the shifts in mutation probability reported by Harris.

To ask whether there are any sub-trends associated with the flanking bases, I next plotted the slope of the relationship with heterozygosity against GC content for each of the six possible mutation types (figure 5). Excepting $C \rightarrow G$ transversions, all other mutations reveal significant ($r > 0.72$, $p < 0.001$) positive correlations. This means that the mutability of a triplet in higher heterozygosity regions increases with triplet GC content. Moreover, the slopes of the relationships are all very similar, even though the intercepts differ. Together, these trends suggest that recent mutations in Africa, being on average in regions with relatively higher heterozygosity, should be skewed towards the types that become more likely in higher heterozygosity regions. Transitions and transversions show opposing trends but transitions are generally commoner. Assuming the overall trend is dominated by transitions, the higher heterozygosity of Africans should favour C/G mutations, while non-Africans should carry relatively more A/T mutations. Such a pattern agrees well with the patterns reported by Harris and Pritchard, suggesting that these are, at least in part, an emergent property generated by a demographically induced change in genome-wide heterozygosity.

The most obvious alternative model to one based on heterozygosity is the one where mutation rate and mutation type are generally correlated, such that mutation hotspots, which will tend to have elevated heterozygosity, also exhibit a different mutation spectrum compared with cold spots. However, this model fails in the case of non-African humans where the massive loss of heterozygosity that occurred 'out of Africa' should have scrambled or even removed any relationship with heterozygosity. The fact that Africans and non-Africans show almost identical dependencies on heterozygosity therefore argues strongly that heterozygosity is causative rather than reflective.

In terms of possible mechanisms, recent studies of de novo mutations in humans reveal differences in the mutation process between early- and late-replicating DNA [4,26,27]. Specifically, late-replicating DNA has a higher mutation rate and is enriched for clustered mutations. In turn, clustered mutations are generally enriched for transversions [28] and show a different palette of mutations, for example, including more $C \rightarrow G$ and $A \rightarrow T$ mutations and fewer $CAT \rightarrow CGG$, $ACG \rightarrow ATT$ and $GCG \rightarrow GTG$ mutations [4,27]. Together, these patterns show how mutation rate, mutation type and mutation non-independence tend to be linked, plausibly through different polymerases, though *cis*-acting factors are also involved [26].

Replication timing does not directly explain the African–non-African difference. For example, clustered de novo mutations show a large excess of most of the possible $C \rightarrow G$ transversions (fig. 4 in [4]) but Africans and non-Africans differ little. However, the basic concept remains. The HI hypothesis is based on the idea that heterozygous sites become mismatches in heteroduplex DNA formed during synapsis which are then 'repaired' by gene conversion events [29]. Like early- and late-replicating DNA, these gene conversions involve different enzymes [30] so will probably exhibit different

mutation spectra, mutation rates and levels of mutation clustering, variables that are then tied to local heterozygosity and thence to the 'out of Africa' bottleneck. Interestingly, it has recently been shown that mutation spectra do differ with ploidy in yeast [31].

Ideally, one would like to model these processes. Unfortunately, the system still appears too complicated to do this convincingly. As well as the suggested dependence on heterozygosity, mutation rate also varies with many other factors including recombination rate [21], GC content [32], replication timing [4], methylation [33] and other factors [2], some of which may interact with/be related to heterozygosity [20] and others will simply act to add noise. Putting these complexities aside, if heterozygosity does modulate both mutation rate and mutation type, even the simplest scenario where both processes are independent, additive and linear, at the very least requires detailed knowledge of how heterozygosity has changed over time, including the length, depth and timing of the bottleneck, the rate and timing of subsequent expansion and the extent of more recent population mixing and movement. In practice, mutation rate, clustering and type are all likely to be linked in ways that have yet to be resolved. Moreover, local haplotype structure is strongly dependent on the local recombination rate which is itself closely tied to the mismatch repair process [30] and mutation rate [21,34,35]. Add to this the possibility that up to eight mutations can occurring in a single event, with neighbouring mutations often occurring on the same strand [4], and it becomes clear that simple models are unlikely to be meaningful.

An interaction between heterozygosity and both the type and the rate of mutations suggests that DNA sequence evolution is, at the same time, both more complicated than is often assumed yet also, in some sense, more predictable. If local heterozygosity is measured, the resulting value has implications for both the rate and the type of likely mutations. The challenge for future research is to discover the rules, and thereby to generate predictive models for how real sequences change over time.

# 3. Methods

## 3.1. Data

All analyses of modern human sequences were performed on data from Phase 3 of the 1000 genomes project [24], downloaded as composite vcf files (available from ftp://ftp.1000genomes.ebi.ac.uk/vol1/ftp/release/20130502/). These comprise low coverage genome sequences for 2504 individuals drawn from 26 modern human populations spread across five main geographical regions: Africa (seven populations), Europe (five populations), Central Southern Asia (five populations), East Asia (five populations) and the Americas (four populations).

## 3.2. Analyses

All analyses were conducted using custom scripts written in C++, illustrated in electronic supplementary material, S1. All statistical analyses were conducted in R v. 3.3.0 (https://cran.r-project.org/). The 1000 g vcf files were searched for variants present in one to five copies where all copies occur in the same major geographical region. At each site, the human reference sequence (hs37d5) was used to infer the mutating ancestral triplet. Flanking heterozygosity was estimated based on all SNPs within 1 kb either side of the mutating triplet, based on all samples from that region (approx. 100 individuals per population = approx. 500 individuals per region, more from Africa, fewer from America). The 2 kb window was chosen based on the estimated size of the gene conversion events documented in yeast, on which HI probably depends [19]. Since the 1000 g data are low coverage, heterozygosity was estimated as expected heterozygosity assuming unlinked loci in Hardy–Weinberg equilibrium. The resulting estimates are conservative in the sense that, despite having the same expected mean, they will have a higher associated variance. Since heterozygosity is only being estimated as a relative value for the last 100 generations or so, this method should be adequate and preferred over the individual-based estimates. Raw count data are available in electronic supplementary material, table S2.

## 3.3. Filtering

While it might be desirable to impose some level of filtering, for example, removing SNPs adjacent to indels because these might be subject to misalignment, I decided not to. My rationale is based on three reasons. First, use of the unfiltered data makes the analysis maximally transparent and avoids

biases that might be introduced by the filtering itself. Second, misalignments are expected to be rare. To test this, I counted how many SNPs were located within five bases of an indel, finding just 0.65%. Third, misalignments will tend to add modest amounts of statistical noise to the analysis, making it conservative, but seem highly unlikely to generate trends of the kind I report.

Data accessibility. I have provided an annotated script of the C++ code I used to extract mutating triplet counts from the 1000 genome data and the raw counts I obtained, both as electronic supplementary material.

Competing interests. I declare no competing interests.

Funding. This work was unfunded.

Acknowledgements. I thank the editor and two anonymous referees for constructive comments that helped improve the quality and clarity of the paper.

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
