## [Reviewer comments · Royal Society Open Science]

Review History

RSOS-191018.R0 (Original submission)

Review form: Reviewer 1

Is the manuscript scientifically sound in its present form?

No

Are the interpretations and conclusions justified by the results?

No

Is the language acceptable?

Yes

Do you have any ethical concerns with this paper?

No

Have you any concerns about statistical analyses in this paper?

Yes

Recommendation?

Major revision is needed (please make suggestions in comments)

Comments to the Author(s)

In “Flanking heterozygosity influences the relative probability of different base substitutions in humans.”, the author suggests that the differences in mutation rates between populations is generally driven by differing patterns of heterozygosity between these populations. This is an interesting hypothesis, but it is not clear that this paper’s analysis provides sufficient evidence to support it.

The major problem of the proposed analysis is the use of local heterozygosity to predict the occurrence of doubleton mutation without correcting for confounding. Many plausible sources for confounding exist as local mutation rates are affected by recombination rate, GC content, replication timing, methylation, lamination, transcription, DNase hypersensitivity, histone marks and probably other factors I forgot to list. All of these factors affect mutation rate and thus affect the underlying heterozygosity and the occurrence of doubleton mutations. Thus these genomic factors may explain the observed relationship without requiring any causal relationship. The author makes a qualitative argument why some of these factors (replication timing and GC content) but a formal model is necessary to support the claims made here.

Second, the results don’t seem to fit with the HI hypothesis, which suggests that “ heterozygosity and mutation rate may be positively correlated.” The paper reports that some mutation types are indeed more common in regions of high heterozygosity (positive correlation) but others are actually less common in regions of high heterozygosity (negative correlation). It seems, the author should explain this contradiction.

Minor issues:

Using the assumption of unlinked loci to calculate heterozygosity is somewhat questionable. Common variants drive heterozygosity and these common variants are called with high precision in 1000G. Hence it is possible to use the 1000G haplotypes directly to calculate heterozygosity. While the statistical model is described reasonably well, a formal definition (in an appendix) would make sure that the reader understands the technical details.

All scripts used for the data analysis should be available via the authors website or via GitHub or a similar website.

Review form: Reviewer 2

Is the manuscript scientifically sound in its present form?

No

Are the interpretations and conclusions justified by the results?

No

Is the language acceptable?

Yes

Do you have any ethical concerns with this paper?

No

Have you any concerns about statistical analyses in this paper?

No

Recommendation?

Major revision is needed (please make suggestions in comments)

Comments to the Author(s)

Population genetic models typically assume that mutations arise randomly, but genomic data are beginning to reveal the scope of mutation rate heterogeneity in genomes, including causal factors. The author's prior work uncovered a link between local heterozygosity and mutation rate. Here, Amos utilizes data from the 1000G project to address one possible explanation for this trend: that heterozygosity is itself mutagenic.

Overall, I think this paper addresses an important biological question, but I have several major concerns about the analysis and its conclusions.

First, it does not appear that the author imposed any filters on the 1000G dataset prior to embarking on this analysis. This is troublesome, as many putative young variants could be artifacts due to read misalignment in regions harboring indels, rare structural variants, or repetitive elements. What filtering steps were carried out, if any? If none, I think the author needs to include some analyses showing that read depths, allele biases, number of neighboring indels, etc., are comparable in doubletons relative to other sites.

Second, I am not convinced that the 1000G dataset is itself the optimal dataset to use for this analysis. For one, as the author acknowledges, the 1000G data are low coverage, necessitating that many variants are imputed. Additionally, this analysis rests on the implicit assumption that doubletons reflect the recent de novo mutation landscape in an unbiased way. This is not necessarily true for mutations that have persisted for multiple generations in the population (as required for a doubleton). There are now a large number of deeply sequenced trios and families in the public domain, and these resources are arguably better suited to addressing the relationship between de novo mutation rate and local heterozygosity.

Third, I thought the text suffered from a lack of key details in places. For example, is heterozygosity calculated on a per-population basis or aggregated over all samples in the 1000G? What filtering steps were carried out on the 1000G VCF file (see point above)?

Fourth, the choice of window size for measuring heterozygosity is, out of necessity, arbitrary. However, the author's conclusion would be strengthened considerably by demonstrating that the relationship between heterozygosity and mutation rate is robust to the use of alternative window sizes.

Finally, I found the presentation of the HI hypothesis on pages 4-5 confusing. How is this hypothesis distinct from the observation that recombination-based repair of double-strand breaks at meiosis is likely mutagenic (e.g. Arbeithuber, et al., 2015, PNAS 112(7): 2109-2114)? I also find it odd that the well-known relationship between recombination rate and DNA diversity is not referenced in this section of the manuscript.

In addition to these major concerns, I also have several minor comments:

- Figures 1 & 5: would be helpful to include trendlines.
- Page 4, lines 18-20: "There is also evidence tying some variation..." As written, this sentence is a bit vague. Perhaps: "There is also evidence tying some variation in mutation rate to variation in DNA mismatch repair genes."
- Page 6, lines 43-45: The phrase "lends strong support" appears twice in the final sentence of this paragraph.
- Page 7, lines 22: Missing "the": "I next plotted THE slope..."

- Page 9, lines 45-60: I'm not convinced that the 26 1000G populations, including their 3-letter codes, need to be included.

Decision letter (RSOS-191018.R0)

22-Jul-2019

Dear Dr Amos,

The editors assigned to your paper ("Flanking heterozygosity influences the relative probability of different base substitutions in humans.") have now received comments from reviewers.

While both referees find the hypothesis interesting, they raise a large number of criticisms and concerns about the approach and analysis undertaken, and therefore are not convinced that the hypothesis is supported by the work presented. We would like you to revise your paper in accordance with the referees' comments which can be found below (not including confidential reports to the Editor). It will be important to take all of the concerns and criticisms into account, which may require additional analyses and approaches to exploring and answering the many points raised by the reviewers. Please note this decision does not guarantee eventual acceptance and the paper once revised will be reviewed again.

Please submit a copy of your revised paper before 14-Aug-2019. Please note that the revision deadline will expire at 00.00am on this date. If we do not hear from you within this time then it will be assumed that the paper has been withdrawn. In exceptional circumstances, extensions may be possible if agreed with the Editorial Office in advance. We do not allow multiple rounds of revision so we urge you to make every effort to fully address all of the comments at this stage. If deemed necessary by the Editors, your manuscript will be sent back to one or more of the original reviewers for assessment. If the original reviewers are not available, we may invite new reviewers.

- Data accessibility

If you wish to submit your supporting data or code to Dryad (<http://datadryad.org/>), or modify your current submission to dryad, please use the following link:
<http://datadryad.org/submit?journalID=RSOS&manu=RSOS-191018>

- Competing interests

- Authors' contributions

- Acknowledgements

- Funding statement

Kind regards,

Andrew Dunn

on behalf of Professor Peter Visscher (Associate Editor) and Steve Brown (Subject Editor)
openscience@royalsociety.org

Comments to Author:

Reviewers' Comments to Author:

Reviewer: 1

Comments to the Author(s)

In "Flanking heterozygosity influences the relative probability of different base substitutions in humans.", the author suggests that the differences in mutation rates between populations is generally driven by differing patterns of heterozygosity between these populations. This is an interesting hypothesis, but it is not clear that this paper's analysis provides sufficient evidence to support it.

The major problem of the proposed analysis is the use of local heterozygosity to predict the occurrence of doubleton mutation without correcting for confounding. Many plausible sources for confounding exist as local mutation rates are affected by recombination rate, GC content, replication timing, methylation, lamination, transcription, DNase hypersensitivity, histone marks and probably other factors I forgot to list. All of these factors affect mutation rate and thus affect the underlying heterozygosity and the occurrence of doubleton mutations. Thus these genomic factors may explain the observed relationship without requiring any causal relationship. The author makes a qualitative argument why some of these factors (replication timing and GC content) but a formal model is necessary to support the claims made here.

Second, the results don't seem to fit with the HI hypothesis, which suggests that "heterozygosity and mutation rate may be positively correlated." The paper reports that some mutation types are indeed more common in regions of high heterozygosity (positive correlation) but others are actually less common in regions of high heterozygosity (negative correlation). It seems, the author should explain this contradiction.

Minor issues:

Using the assumption of unlinked loci to calculate heterozygosity is somewhat questionable. Common variants drive heterozygosity and these common variants are called with high precision in 1000G. Hence it is possible to use the 1000G haplotypes directly to calculate heterozygosity. While the statistical model is described reasonably well, a formal definition (in an appendix) would make sure that the reader understands the technical details.

All scripts used for the data analysis should be available via the authors website or via GitHub or a similar website.

Reviewer: 2

Comments to the Author(s)

Population genetic models typically assume that mutations arise randomly, but genomic data are beginning to reveal the scope of mutation rate heterogeneity in genomes, including causal factors. The author's prior work uncovered a link between local heterozygosity and mutation rate. Here, Amos utilizes data from the 1000G project to address one possible explanation for this trend: that heterozygosity is itself mutagenic.

Overall, I think this paper addresses an important biological question, but I have several major concerns about the analysis and its conclusions.

First, it does not appear that the author imposed any filters on the 1000G dataset prior to embarking on this analysis. This is troublesome, as many putative young variants could be artifacts due to read misalignment in regions harboring indels, rare structural variants, or repetitive elements. What filtering steps were carried out, if any? If none, I think the author needs to include some analyses showing that read depths, allele biases, number of neighboring indels, etc., are comparable in doubletons relative to other sites.

Second, I am not convinced that the 1000G dataset is itself the optimal dataset to use for this analysis. For one, as the author acknowledges, the 1000G data are low coverage, necessitating that many variants are imputed. Additionally, this analysis rests on the implicit assumption that doubletons reflect the recent de novo mutation landscape in an unbiased way. This is not necessarily true for mutations that have persisted for multiple generations in the population (as required for a doubleton). There are now a large number of deeply sequenced trios and families in the public domain, and these resources are arguably better suited to addressing the relationship between de novo mutation rate and local heterozygosity.

Third, I thought the text suffered from a lack of key details in places. For example, is heterozygosity calculated on a per-population basis or aggregated over all samples in the 1000G? What filtering steps were carried out on the 1000G VCF file (see point above)?

Fourth, the choice of window size for measuring heterozygosity is, out of necessity, arbitrary. However, the author's conclusion would be strengthened considerably by demonstrating that the relationship between heterozygosity and mutation rate is robust to the use of alternative window sizes.

Finally, I found the presentation of the HI hypothesis on pages 4-5 confusing. How is this hypothesis distinct from the observation that recombination-based repair of double-strand breaks at meiosis is likely mutagenic (e.g. Arbeithuber, et al., 2015, PNAS 112(7): 2109-2114)? I also find it odd that the well-known relationship between recombination rate and DNA diversity is not referenced in this section of the manuscript.

In addition to these major concerns, I also have several minor comments:

- Figures 1 & 5: would be helpful to include trendlines.
- Page 4, lines 18-20: "There is also evidence tying some variation..." As written, this sentence is a bit vague. Perhaps: "There is also evidence tying some variation in mutation rate to variation in DNA mismatch repair genes."
- Page 6, lines 43-45: The phrase "lends strong support" appears twice in the final sentence of this paragraph.
- Page 7, lines 22: Missing "the": "I next plotted THE slope..."
- Page 9, lines 45-60: I'm not convinced that the 26 1000G populations, including their 3-letter codes, need to be included.

Author's Response to Decision Letter for (RSOS-191018.R0)

See Appendix A.

Decision letter (RSOS-191018.R1)

28-Aug-2019

Dear Dr Amos:

On behalf of the Editors, I am pleased to inform you that your Manuscript RSOS-191018.R1 entitled "Flanking heterozygosity influences the relative probability of different base substitutions in humans." has been accepted for publication in Royal Society Open Science subject to minor revision in accordance with the Editors' suggestions.

The Associate and Subject Editor have recommended publication, but also suggest some minor revisions to your manuscript. Note that the submission pdf contains the manuscript twice, and the first set of Figures 1 & 3 do not have panel labelling of the triplets whereas the second set does. Also, Figures 1 & 2 and 3 & 4 have the same title, so one of those titles needs changing. Please address these minor comments and revise your manuscript.

- Ethics statement

- Data accessibility

<http://datadryad.org/submit?journalID=RSOS&manu=RSOS-191018.R1>

- Competing interests

- Authors' contributions

- Acknowledgements

- Funding statement

Because the schedule for publication is very tight, it is a condition of publication that you submit the revised version of your manuscript before 06-Sep-2019. Please note that the revision deadline will expire at 00.00am on this date. If you do not think you will be able to meet this date please let me know immediately.

Supplementary files will be published alongside the paper on the journal website and posted on

the online figshare repository (<https://figshare.com>). The heading and legend provided for each supplementary file during the submission process will be used to create the figshare page, so please ensure these are accurate and informative so that your files can be found in searches. Files on figshare will be made available approximately one week before the accompanying article so that the supplementary material can be attributed a unique DOI.

on behalf of Professor Peter Visscher (Associate Editor) and Steve Brown (Subject Editor)
openscience@royalsociety.org

Author's Response to Decision Letter for (RSOS-191018.R1)

I have made the changes requested. Thank you. Bill.

Decision letter (RSOS-191018.R2)

30-Aug-2019

Dear Dr Amos,

I am pleased to inform you that your manuscript entitled "Flanking heterozygosity influences the relative probability of different base substitutions in humans." is now accepted for publication in Royal Society Open Science.

Kind regards,
Lianne Parkhouse
Editorial Coordinator
Royal Society Open Science
openscience@royalsociety.org

on behalf of Professor Peter Visscher (Associate Editor) and Steve Brown (Subject Editor)
openscience@royalsociety.org

Appendix A

Dear Editor,

I am grateful for the Reviewers' comments, many of which were very helpful in guiding me towards better clarity and presentation. I have now conducted extensive revision both of the text and of the results. Specifically, I have rewritten my code from scratch and repeated all analyses, extending these to variants present in one to five copies), and explored different window sizes. During this process I discovered two minor issues. First I had inadvertently excluded exonic sequences. The impact was trivial but has now been corrected. Second, I had previously required each doubleton to be present only in one population, even though the unit used in the analyses is major geographic region. I now use major geographic region for both classifications. I also shifted from taking the first 30 heterozygosity bins of 50, above which numbers became scarce, to multiplying heterozygosity by a factor that maximises the evenness of counts across the 0 – 30 range, combining 30+ into a single class. While the overall picture remains little changed, in that triplet mutability is strongly and ubiquitously dependent in heterozygosity, there has been a small but noticeable shift in the relatively mutabilities of some triplets. Part of this is due to the relative weighting of low as opposed to zero heterozygosity regions, hence the shift in some of the slopes and even the direction of some slopes. I now include plots so that the reader can see directly how much scatter and how strong the relationships are for all mutation-triplet-population combinations.

Below are detailed responses to each and every substantive comment made by the two reviewers. Throughout, my comments are in red and reviewer comments are unedited in black:

Reviewer: 1

Comments to the Author(s)

In "Flanking heterozygosity influences the relative probability of different base substitutions in humans.", the author suggests that the differences in mutation rates between populations is generally driven by differing patterns of heterozygosity between these populations. This is an interesting hypothesis, but it is not clear that this paper's analysis provides sufficient evidence to support it.

The major problem of the proposed analysis is the use of local heterozygosity to predict the occurrence of doubleton mutation without correcting for confounding. Many plausible sources for confounding exist as local mutation rates are affected by recombination rate, GC content, replication timing, methylation, lamination, transcription, DNase hypersensitivity, histone marks and probably other factors I forgot to list. All of these factors affect mutation rate and thus affect the underlying heterozygosity and the occurrence of doubleton mutations. Thus these genomic factors may explain the observed relationship without requiring any causal relationship.

- **Response:** There are indeed many causes of variation in mutation rate and these will, over time, lead to variation in heterozygosity. However, I test for a relationship between heterozygosity and mutation **type** (not **rate**, as the Reviewer's wording suggests). My primary finding is that mutation **type** varies systematically with local heterozygosity and this finding remains, regardless of how / why heterozygosity varies. I then point out that the relationship between mutation type and heterozygosity is indistinguishable between Africans and non-Africans. Since Africans have both much higher heterozygosity and different mutation spectra, the logical conclusion is that loss of heterozygosity 'out of Africa' played an important role in driving changes in which types of mutation are more likely. I try to make this clearer in the revised manuscript, including an extra sentence in the abstract.

The author makes a qualitative argument why some of these factors (replication timing and GC content) but a formal model is necessary to support the claims made here.

- **Response:** I do not understand this request and would welcome Editorial guidance. I believe I show clearly that mutation type and heterozygosity are linked. I then speculate about a plausible mechanism, supported by published work showing that early and late-replicating DNA exhibit different mutation spectra. I do not understand what 'modelling' I could do, formal or otherwise, that would shed further light. I do, however, discuss why modelling the whole process is not really possible at this time.

Second, the results don't seem to fit with the HI hypothesis, which suggests that " heterozygosity and mutation rate may be positively correlated." The paper reports that some mutation types are indeed more

common in regions of high heterozygosity (positive correlation) but others are actually less common in regions of high heterozygosity (negative correlation). It seems, the author should explain this contradiction.

- **Response:** I have added a sentence to make this clearer in the revised text. The frequencies of all possible mutation types must sum to one, so if some types increase with heterozygosity, others will inevitably decrease. This is a direct consequence of how the data are presented and there is no contradiction.

Minor issues:

Using the assumption of unlinked loci to calculate heterozygosity is somewhat questionable. Common variants drive heterozygosity and these common variants are called with high precision in 1000G. Hence it is possible to use the 1000G haplotypes directly to calculate heterozygosity.

While the statistical model is described reasonably well, a formal definition (in an appendix) would make sure that the reader understands the technical details.

- **Response:** I have made this clearer in the revised methods. Individual genotypes in 1000g are not very reliable because of imputation and this is particularly true for inferences about phase (necessary for inferring haplotypes). What **are** reliable are allele frequencies, since these are based on ~200 allele calls per population. Linkage will increase the variance but should not change the mean of the estimate. As such, the assumption of no linkage will, if anything, add noise and cause the strength of the relationship to be under-estimated. As above, I do not understand what is meant or requested by a 'formal definition' but will do my best to provide what is needed when it is explained.

All scripts used for the data analysis should be available via the authors website or via GitHub or a similar website.

- **Response:** I have added outline annotated scripts as supplementary material.

Reviewer: 2

Comments to the Author(s)

Population genetic models typically assume that mutations arise randomly, but genomic data are beginning to reveal the scope of mutation rate heterogeneity in genomes, including causal factors. The author's prior work uncovered a link between local heterozygosity and mutation rate. Here, Amos utilizes data from the 1000G project to address one possible explanation for this trend: that heterozygosity is itself mutagenic.

Overall, I think this paper addresses an important biological question, but I have several major concerns about the analysis and its conclusions.

First, it does not appear that the author imposed any filters on the 1000G dataset prior to embarking on this analysis. This is troublesome, as many putative young variants could be artifacts due to read misalignment in regions harboring indels, rare structural variants, or repetitive elements. What filtering steps were carried out, if any? If none, I think the author needs to include some analyses showing that read depths, allele biases, number of neighboring indels, etc., are comparable in doubletons relative to other sites.

- **Response:** I did not impose filters for three important reasons. First, all the issues listed will tend to add noise to the analysis and thereby reduce the strength of any trends I find and make my analysis conservative. Thus, while filtering might help strengthen the trends, the effect is likely very weak because most issues are rare (in response to this comment I counted how many SNPs lie within 5 bases of an indel, finding just 0.6%). Crucially, failure to filter should not create false trends. Second, I am not a bioinformatician and hence trust the large body of technically gifted people who curated the 1000g data to do a far better job than me at minimising these problems. Third, pragmatically, many are sceptical of my work in this area and my experience is that any mention of extensive filtering will only increase this scepticism. I have added a paragraph to the methods to explain this.

Second, I am not convinced that the 1000G dataset is itself the optimal dataset to use for this analysis. For one, as the author acknowledges, the 1000G data are low coverage, necessitating that many variants are imputed. Additionally, this analysis rests on the implicit assumption that doubletons reflect the recent de novo mutation landscape in an unbiased way. This is not necessarily true for mutations that have persisted for multiple generations in the population (as required for a doubleton). There are now a large number of deeply sequenced trios and families in the public domain,

- **Response:** There are two possible approaches that I now discuss explicitly in the revised text. One is to use trios and to ask about the relationship between verified *de novo* mutations and their context. However, in my experience, the availability of high quality trio data is not as great as the Reviewer implies. Even the massive Icelandic dataset (which I explored using but this required a several month visit to Iceland that I could not accommodate) yields 'only' 17,000 mutations. This seems a lot, but is still borderline in terms of testing for subtle patterns across many triplet mutation classes. The alternative, which I choose, is to exploit the fact that the overwhelming majority of rare variants in a population reflect recent mutations. Here I sacrifice the precision of the trio approach for the vastly greater numbers of sites that exist in population samples. In the population approach, **individual** heterozygosity is not useful because the variants are tens or hundreds of generations old. As such, the best estimate of the heterozygosity surrounding them when the mutation occurred is the population average. The 1000g data offer the best balance between number of populations sampled and the possibility of estimating population heterozygosity with high precision (despite being low coverage, each population is represented by ~200 allele calls). I make this choice clearer in the revised text.

and these resources are arguably better suited to addressing the relationship between *de novo* mutation rate and local heterozygosity.

- **Response:** The main focus of the current paper is to test for a link between heterozygosity and mutation **type**, not **rate**. In fact, in my experience both approaches are likely valid but both have drawbacks. When I analysed population data, I find that recent mutations are highly significantly more likely in regions that have elevated heterozygosity, even relative to closely related populations from the same geographic region.

Third, I thought the text suffered from a lack of key details in places. For example, is heterozygosity calculated on a per-population basis or aggregated over all samples in the 1000G? What filtering steps were carried out on the 1000G VCF file (see point above)?

- **Response:** More detail has been added, as requested.

Fourth, the choice of window size for measuring heterozygosity is, out of necessity, arbitrary. However, the author's conclusion would be strengthened considerably by demonstrating that the relationship between heterozygosity and mutation rate is robust to the use of alternative window sizes.

- **Response:** The choice is not completely arbitrary. The gene conversion events in yeast are described as being of the order of 1-2Kb in length and this is what I based my window size on. I have made this clear in the revised text. I also analysed a window size 10X larger and found effectively identical results. This is not surprising, since at the kilobase level there is a high degree of autocorrelation along a chromosome.

Finally, I found the presentation of the HI hypothesis on pages 4-5 confusing. How is this hypothesis distinct from the observation that recombination-based repair of double-strand breaks at meiosis is likely mutagenic (e.g. Arbeithuber, et al., 2015, PNAS 112(7): 2109-2114)?

- **Response:** The HI hypothesis greatly predates this work, dating back to a study I did with David Rubinsztein in 1995 and specifically suggests that mutation rate is higher at and around heterozygous sites. The mechanism has been empirically documented in yeast and involves gene conversion events attracted to mismatches that occur when heterozygous sites are included in heteroduplex DNA formed during synapsis. Since these gene conversion events can be resolved by recombination, I have long speculated that this should link mutation rate and recombination rate as is observed. As the Reviewer correctly points out, DNA synthesis associated with recombination will also increase local mutation rate, and this is true regardless of whether the event was initiated by the presence of a heterozygous site. From the population genetic perspective, it is the link to heterozygosity that is most interesting (and heretical!) because this ties mutation rate to demographic changes. I have extended the referencing to this effect, including to the paper suggested by the Reviewer.

I also find it odd that the well-known relationship between recombination rate and DNA diversity is not referenced in this section of the manuscript.

- **Response:** see above. Note, I was not trying to be exhaustive about HI because the main focus of the paper is the link between heterozygosity and mutation type, a message that appears not to be as clear as I hoped! More references have been added along the lines suggested.

In addition to these major concerns, I also have several minor comments:

- Figures 1 & 5: would be helpful to include trendlines.

- **Response:** in the revised text new figures are presented where I think trendlines would decrease legibility substantially. I think the extra data and ability to see many graphs side-by-side negates the need for trendlines.

- Page 4, lines 18-20: "There is also evidence tying some variation..." As written, this sentence is a bit vague. Perhaps: "There is also evidence tying some variation in mutation rate to variation in DNA mismatch repair genes."

- Page 6, lines 43-45: The phrase "lends strong support" appears twice in the final sentence of this paragraph.

- Page 7, lines 22: Missing "the": "I next plotted THE slope..."

- Page 9, lines 45-60: I'm not convinced that the 26 1000G populations, including their 3-letter codes, need to be included.

- **Response:** all corrected, thank you.